# Assessing Cultural, Religious, and Trauma Influences in Human-Animal Interactions for Effective Animal-Assisted Counseling

**DOI:** 10.3390/ani14172496

**Published:** 2024-08-28

**Authors:** Jordan Jalen Evans

**Affiliations:** Department of Counseling and Educational Psychology, Texas A&M University-Corpus Christi, Corpus Christi, TX 78412, USA; drjaelpc@outlook.com

**Keywords:** animal-assisted counseling, competencies, human–animal interaction, human–animal bond, cultural humility, race, animal-related trauma, animal welfare

## Abstract

**Simple Summary:**

AAC involves a trained mental health practitioner, a therapy animal, and a client participating in goal-oriented interventions. These sessions, characterized by human–animal interactions, aim to strengthen the human–animal bond when beneficial to both client and animal. Through this bond, clients may experience reduced mental health symptoms, improved communication skills, and a stronger therapeutic relationship. While research suggests that human–animal interactions can enhance rapport in counseling, it is crucial to recognize that clients and practitioners may have varying perceptions of these interactions. Historically, animals have been used in aggressive and violent practices within communities of color, and, across different cultures and religions, animals may be revered, forbidden, or slaughtered. Therefore, animal-assisted practitioners must address trauma, race, culture, and religion as they relate to animals before introducing AAC. This article aims to explore how AAC practitioners can prevent harm to the human–animal bond by practicing cultural humility.

**Abstract:**

The purpose of this manuscript is to enhance the understanding of how racial, social, and cultural factors influence animal-assisted counseling (AAC). As AAC gains popularity, there is an increasing need for clinicians to practice cultural humility and awareness. While AAC has proven beneficial, clinicians must consider the diverse cultural, religious, and trauma-related perceptions of animals. The American Counseling Association (ACA) has established AAC competencies that highlight the importance of understanding these social and cultural factors, assessing past animal-related trauma, and evaluating client suitability for AAC in the United States. Similarly, in 2018, the International Association of Human-Animal Interactions Organizations (IAHAIO) and, in 2024, the Association of Animal-Assisted Intervention Professionals (AAAIP) set standards for competencies related to clients’ cultural backgrounds, trauma, and historical oppressions related to certain species. By addressing these considerations, clinicians can better promote and protect the welfare of both clients and therapy animals. While these organizations generally emphasize ethical standards, professional guidelines, and safeguarding client–animal relationships, this manuscript advocates for a more robust examination of cultural, racial, and societal factors in the use of AAC. This includes not only recognizing the ethical implications but also understanding how diverse backgrounds and access disparities shape the effectiveness, acceptability, and accessibility of AAC interventions. This approach integrates culturally responsive practices and promotes a deeper exploration of how race, culture, religion, and societal factors influence human–animal relationships.

## 1. Introduction

Due to the important and necessary topics discussed in this article, the author finds that it would be beneficial to be authentic in introducing herself. This author is an animal-assisted counselor, a bi-racial, cisgender woman who is African and Mexican American. This information should not be generalized but rather used to help practitioners build a foundational understanding of cultural beliefs and associated traumas related to animals in various populations.

While many AAC articles discuss competencies and animal welfare, they often overlook considerations of culture, religion, and race. Additionally, there is a significant gap in research concerning AAC and people of color. This is a gap in most research in America, in which Black and Latino populations are underrepresented when it comes to mental health and health needs [1]. This is an issue as people of color are the global majority. The lack of research on this issue may stem from socioeconomic barriers, limited access to mental health services and specialized treatments like AAC in some communities, and the high cost of AAC training for clinicians. For example, one study found that during COVID-19, white people were more likely to receive mental health treatment due to systemic privileges, though communities of color, specifically Black, Hispanic, and Asian people, showed high levels of need and were more likely to face long-term challenges [2].

Animal-assisted counseling (AAC) and other animal-assisted interventions are becoming increasingly popular, highlighting the need to consider ethical and cultural implications to protect the client–animal relationship. As mentioned above, while AAC has demonstrated therapeutic benefits, existing literature often overlooks cultural, religious, and racial factors, with a notable gap in research involving people of color, who are the global majority in America. Social and cultural factors, religion, historical injustices, experiences, and trauma play a significant role in animal-assisted interventions (AAI) and can greatly impact human–animal interactions. For example, in some communities, animals are seen primarily as companions, while in others, they may be viewed more as utilitarian or sacred. Understanding these factors is crucial for implementing effective and culturally sensitive practices.

AAC practitioners must consider ethical and cultural factors to protect the client–animal relationship. Culture, race, and social influences significantly shape human–animal interactions, affecting how different societies view, treat, and coexist with animals [3]. In many cultures, animals hold symbolic meanings, often tied to religious beliefs, myths, and traditions. For example, cows are considered sacred in Hindu culture, leading to their protection and reverence, while in some Indigenous cultures, animals like eagles or wolves are revered as spiritual guides or symbols of strength [4,5]. Dietary customs also vary widely, with certain animals being consumed as a staple in some cultures while being taboo in others. Additionally, cultural practices influence the roles animals play, such as using horses for transportation, dogs for herding or companionship, and birds in ceremonial activities [6]. Urbanization and modernization have further altered these interactions, with animals increasingly being kept as pets in some cultures, while in others, they continue to serve as essential sources of labor or sustenance. These cultural frameworks shape the ethics, laws, and everyday practices surrounding human–animal relationships, creating a diverse and complex tapestry of interactions worldwide.

Animal-assisted counseling is defined as the inclusion of the human–animal bond throughout the treatment process by a mental health practitioner [7]. The human–animal bond (HAB) is characterized as a mutually beneficial relationship [8], meaning that both the animals and the humans involved in the counseling process benefit from their interactions (HAI). Animal-assisted practitioners are responsible for promoting positive, beneficial interactions between clients and therapy animals, while prioritizing the welfare of both humans and animals.

According to the American Counseling Association (ACA), animal-assisted practitioners must understand how cultural factors influence human–animal interactions, ensure animal welfare, assess for past animal-related trauma, and evaluate the suitability of both the client and the animal for AAC [9]. Practitioners are also required to avoid causing harm to clients, honor diversity and cultural backgrounds, and protect the integrity of the counselor–client relationship [10]. Furthermore, the Association of Animal-Assisted Intervention Professionals (AAAIP) has a set of special considerations for best practices within the profession, which states that practitioners should identify cultural and historical oppressions related to various animal species, recognize fears in work with animals, and determine factors related to social and cultural implication in human–animal interactions [11]. Therefore, it is recommended that practitioners consider the impact of various experiences, such as racism-related stressors and cultural factors like religion, on clients’ perceptions of the human–animal bond [12].

The human–animal bond is formed through interactions between the client and the animal, with the strength of this relationship depending on the nature of these exchanges [7]. Although research is limited, it is understood that this bond can be influenced by various factors and clients’ past experiences (e.g., a client who screams during an interaction with a therapy dog due to a traumatic event involving a dog). Conversely, the bond can be strengthened through positive experiences and cultural perspectives related to a particular species.

As illustrated in Figure 1, the trained AAC practitioner plays a key role in the therapeutic relationship between the client and the therapy animal. The human–animal bond is intended to be a positive relationship, where both the animal and the human benefit from their interactions during sessions. However, if practitioners are unprepared or unaware of their client’s needs and readiness for AAC, it could lead to potential harm to the client, the animal, and the clinician. The International Association of Human-Animal Interaction Organizations (IAHAIO) suggests that, to ensure client well-being, practitioners should seek specialized training, assess the appropriateness of AAC, evaluate risks, understand species and client perceptions, and adjust for cultural factors that may not align with typical AAC procedures [13]. Intensive training specific to working with therapy animals in a mental health setting is recommended [14]. In a two-year study, Hartwig and Evans found that aligning a training program with AAC competencies, combined with in-person supervision and clinical experience, significantly improved the knowledge, skills, and attitudes necessary for AAC practitioners and their therapy animals to succeed with clients [15]. Through such training, practitioners can learn to implement protective measures for themselves, their clients, and their animals [13].

## 2. Background

Animal-assisted counseling (AAC) is a therapeutic approach that involves the collaboration of trained animals, trained animal-assisted practitioners, and clients with specific mental health treatment goals. Research has demonstrated that AAC offers numerous benefits, including strengthening the therapeutic relationship, enhancing the human–animal bond, and promoting positive changes in mental health and behavior [16,17]. While AAC can effectively help clients address issues such as trauma, abuse, and other concerns, it is crucial for practitioners to assess clients’ histories before introducing them to therapy animals [18].

In the counseling profession within the United States, practitioners are required to adhere to a set of ethical codes [10]. Additionally, the American Counseling Association (ACA) provides specific guidelines for those incorporating therapy animals into treatment [9]. Both sets of standards emphasize the importance of understanding each client’s cultural background, minimizing risks, evaluating their history, and ensuring that no harm comes to the client. This paper will provide a brief overview of the roles animals play in religion, culture, and experiences of animal-related trauma. Practitioners must assess their clients’ individual experiences while also being mindful of their own cultural and personal biases.

### Race, Trauma, Religion, and Culture

To ensure ethical practice and maximize the benefits of the human–animal bond, the American Counseling Association (ACA), the International Association of Human-Animal Interactions Organizations (IAHAIO), and the Association of Animal-Assisted Intervention Professionals (AAAIP) urge practitioners to consider various cultural factors when engaging in AAC [9,11,13]. Practitioners should stay informed on current literature related to AAC and develop an understanding of cultural perceptions regarding human–animal relationships, particularly concerning the species involved [9].

While animal-assisted counseling (AAC) has demonstrated benefits and is gaining popularity, it is crucial for practitioners to be mindful of the diverse perceptions of animals shaped by culture, religion, and experiences with animal-related trauma. Race plays a significant role in shaping human–animal interactions, often influencing the way individuals and communities perceive, treat, and relate to animals. Cultural backgrounds tied to race can dictate attitudes towards certain animals, such as viewing them as sacred, pests, companions, or food sources. For instance, in some cultures, which are often associated with specific racial or ethnic groups, certain animals like cows or dogs might be revered and protected, while in others, they may be utilized for labor or consumption. Additionally, socio-economic disparities often linked to race can impact access to resources like pet ownership, veterinary care, and wildlife conservation efforts. These interactions are further shaped by historical legacies of colonialism and racism. Thus, race intersects with cultural, economic, and historical factors to deeply influence how different groups of people engage with animals.

Social injustices and racism throughout U.S. history have led to various harmful incidents involving animals. During the era of slavery and the Civil Rights Movement, police dogs were used to control and intimidate African Americans, contributing to trauma and a mistrust of dogs in some communities. For example, during the era of the Underground Railroad, the spiritual song “Wade in the Water” was used by Harriet Tubman as a coded message, instructing escaping slaves to enter water to avoid detection by dogs trained to track them. Similarly, during the Civil Rights Era, police dogs were frequently used to intimidate and, in some cases, attack African Americans protesting for their rights [19].

Throughout history, dogs have been used in military and police roles, often leading to their perception as tools of oppression or aggression in certain contexts. In 1991, a lawsuit was filed against the Los Angeles Police Department (LAPD) after it was revealed that over 900 people, predominantly Black and Latino, had been bitten by LAPD police dogs within a three-year span [20]. This lawsuit alleged that the dogs were deployed in non-violent situations by handlers who were inadequately trained and supervised, raising concerns about racial discrimination and unconstitutional practices. The lawsuit demanded that the K9 unit be suspended until both the dogs and their handlers were properly selected and trained [20].

Following the 2014 Ferguson unrest, the Department of Justice investigated the Ferguson Police Department in 2015, uncovering numerous instances where police dogs were used to bite low-level offenders or innocent individuals [21]. Notably, every reported victim was an African American male, either a school-aged boy or an adult [21]. These historical and ongoing incidents highlight the importance of understanding animal-related trauma, especially when working with dogs, which are a common therapy animal in AAC in the United States.

Furthermore, attention to specific species and cultural perceptions is a necessity [11]. For example, canines are considered “unclean” in Islam, while cats are more commonly accepted in homes and family settings [22]. Cats are generally accepted and were sacred in Egyptian history but still have lingering reputations of superstition associated with witches and witchcraft in the United States and in some religions [23]. In the United States, cats are second to dogs in pet ownership [24]. Cows are revered as “sacred” in Indian culture, whereas cows are accepted for consumption and pigs are forbidden in Islam [25,26]. Jewish dietary laws also prohibit the consumption of pork [26]. In the United States, cows are primarily seen as livestock for consumption rather than as pets or companions. In regions like Europe, in France, Germany, and Poland, and Central Mexico, rabbits are commonly eaten, whereas in the United States, they are often viewed as pets and eating them can be socially frowned upon.

In Western cultures, dogs are often seen as loyal companions and family members. Regarding dog bites and trauma in the United States, white people are most often shown to be bitten at a higher rate than Black, Asian, or Indigenous people [27]. White and Hispanic adults are reported to own pets, mostly dogs, in the United States, as well as Americans living in rural areas versus urban areas [24]. Research has discussed that these statistics can be inaccurate and biased due to a lack of people seeking medical care or the inability to seek medical care and underreporting [27]. Additionally, the proportion of pet ownership versus animal bites are going to be shown in the data.

In religions such as Judaism, Christianity, and Islam, animals are often seen as resources for human benefit, yet these faiths also promote compassion, respect, and love for animals, with the belief that all creatures are known to God or Allah [28]. Horses were often central to the colonial expansion and exploitation of Indigenous lands and peoples, but archeologists have found that the domestication of horses within Indigenous cultures could have begun as early as the 1600s [29]. Before European colonizers arrived, Indigenous persons were most likely integrating horses into their daily lives. In modern day, many Indigenous communities continue to share a strong bond with horses and incorporate them into traditions and ceremonies [30]. Additionally, within Indigenous history, wolves were used for hunting and protection, eventually becoming domesticated and regarded as family companions. However, in contemporary times, Indigenous reservations face challenges like underfunding, lack of education, and poverty, leading to issues such as large populations of stray dogs and an increase in dog attacks [31].

This information cannot be generalized for either animal species nor humans and is meant to assist practitioners to have a foundation of understanding cultural beliefs and associated traumas with various population involving animals. Therefore, animal-assisted practitioners should be aware of their clients’ unique beliefs through intercultural communication, have cultural self-awareness, all while also taking into consideration what would be safe for both the animal and the client.

## 3. Considerations

Trained animal-assisted counseling (AAC) practitioners should respect clients’ autonomy in deciding whether to participate in AAC. Just as animals in AAC have the choice to engage in sessions, clients also have the right to choose their involvement. Practitioners must avoid pressuring clients into interactions with therapy animals to protect the well-being of all parties involved. Problems in the human–animal bond can occur when practitioners do not respect clients’ autonomy, often stemming from the misconception that animals appeal to everyone. This assumption can be harmful, particularly if it disregards clients’ cultural perspectives and past traumatic experiences.

A client may agree to participate in AAC but, without assessment, may be hiding a history of trauma or various perceptions of animals. Even if clients agree to participate, their underlying fears can cause significant anxiety and stress, which may undermine the effectiveness of the therapy. A strong therapeutic relationship relies on trust. Fears or phobias can create barriers to developing this trust between the client, the therapist, and the therapy animal. The presence of a therapy animal might disrupt the therapeutic process if the client’s fears are not addressed properly, leading to discomfort or avoidance. A fearful or anxious client may unintentionally cause stress for the therapy animal. Animals can pick up on human emotions, and a negative interaction could impact their behavior and well-being. In extreme cases, clients might act out their fears in a way that could be harmful to the animal or themselves, making it essential to manage these fears carefully. If a client’s fears are not managed, the therapeutic benefits of AAC may be diminished. Effective AAC relies on positive interactions, and fear can inhibit this process. Though addressing fears and phobias can be a goal within the therapy itself, this should be done with caution. For clients with specific animal-related phobias, the intervention might need to focus on gradually overcoming these fears. The following discussion will address considerations for preventing such issues and ensuring respectful and effective AAC practices.

When a client expresses interest in participating in animal-assisted counseling (AAC), it is crucial for practitioners to assess the client’s suitability for this modality, considering the species of the therapy animal (see Figure 2). This assessment is based on a survey where AAC practitioners rated the importance of various questions for informed consent [32]. The top-rated questions identified as “extremely important” include (1) history of aggression or abuse, (2) animal fears or phobias, (3) negative experiences with animals, (4) animal/food allergies, (5) interest in participating in AAC, (6) cultural considerations, (7) purpose of AAC, (8) positive experiences with animals, (9) zoonoses, and (10) current or former pets [32].

Assessing client appropriateness is essential to ensure the safety and well-being of both the animal and the client. Practitioners should evaluate the client’s interest in AAC, current and past pet ownership, allergies to animals or foods (including those related to the therapy animal’s diet), negative or traumatic experiences with animals, history of animal abuse or violence, and any religious, spiritual, or cultural beliefs about animals and provide education on roles and definitions of what AAC consists of.

To effectively conduct this assessment, practitioners should discuss the responses with the client and ask clarifying questions to gain a thorough understanding and develop a tailored treatment plan. Practitioners should constantly engage in open dialogue with clients about their fears and other concerns regarding AAC and/or the animal species. This helps in creating a safe and comfortable environment for discussing and addressing these issues. Additionally, an initial session outlining the rules, informed consent, risks, benefits, and addressing any client questions before the first physical meeting with the therapy animal can help clients know what to expect [33].

### Cultural Responsiveness

Cultural responsiveness is an ethical duty of practitioners [10]. Every client seen in therapy has cultural and societal perceptions. Therefore, responding to all clients with respect to their culture and ideas of human–animal interactions is a responsibility to an AAC practitioner. One effective method for exploring culture, race, and other factors impacting the client, practitioner, and human–animal bond is broaching. Broaching typically begins in the initial sessions between the practitioner and client and acknowledges differences between them [34]. This skill should be consistently practiced throughout the counseling relationship to foster intercultural communication and demonstrate cultural humility [35]. Addressing ethnic-cultural topics can enhance client safety, strengthen the therapeutic relationship, and create a more comfortable experience [36]. When working with an animal partner, practitioners must consider how cultural, racial, trauma-related, and other factors influence human–animal interactions and the formation of the human–animal bond, through client engagement, genuine curiosity, and self-assessment.

Animal-assisted practitioners should be aware of their personal, cultural, and racial biases and how these may impact the therapeutic relationship. They should also understand how their therapy animal may be perceived by others and be prepared to assess clients’ sense of safety [9]. Given the triadic nature of the relationship, practitioners need to monitor the therapy animal’s responses to the environment to ensure its welfare [9].

The first session is crucial for ensuring the safety of both the therapy animal and the client before they meet. Assessing client appropriateness, as previously discussed, is vital for determining if a client is suitable for AAC. Additionally, informed consent policies and procedures should be clearly outlined. Based on the 2021 survey, informed consent should address the AAC practitioner’s role, education, and credentials, client health and wellness, sanitization procedures, animal illness protocols, and risks of animal-transmitted diseases [32]. It should also cover the therapy animal’s rights, the benefits of the human–animal bond and AAC, the animal’s specific body language and communication, behavioral boundaries (e.g., no kicking or picking up the animal), and risk prevention. These elements are often overlooked by practitioners not specifically trained in AAC [7,21]. All forms should be reviewed and signed by the client and, if applicable, their guardian(s). During the initial session, AAC practitioners should use questions to further understand the client’s beliefs and comfort level regarding working with an animal in counseling. This type of evaluation process should be utilized for all clients regardless of race or cultural background. An overview of the suggested evaluation process is provided below.


**Animal-Assisted Counseling Evaluation Process**
Practitioner provides education on the integration of animal-assisted counseling in their practice. Client decides whether they are interested in AAC. (If not interested, the practitioner does not force the client nor bring the client into session.) If client expresses interest in AAC, the following can be utilized to guide the initial session (avoid bringing animal to initial session):Assess appropriateness for AACAssess following the guide in Figure 2. Utilize broaching techniques to acknowledge differences and ask questions to gain a deeper understanding of client’s perspective. Get more information on any instances of trauma and cultural beliefs of animals, and assess client’s preferences and comfortability with various species, interventions, and any past work to mend relationships with animals.If appropriate for AAC with your therapy animal, provide informed consentProvide client with information on what animal-assisted counseling is and how the therapy animal will be involved in session.Within the informed consent, provide policies and procedures. Make sure to go in depth with what the role of the AAC practitioner is and the training received, health considerations, the therapy animal’s rights and ways you will ensure welfare, behaviors and communications of the specific therapy animal that is working in session, benefits and risks of AAC, and ways to prevent these risks. 

After gathering this information, AAC practitioners can then alter their treatment for what is best for the client, themselves, and the therapy animal. This may mean choosing to not have the client participate in AAC or deciding to engage in AAC. This could look like tailoring interventions for the client’s specific needs, choosing an alternative therapy animal if there is access to multiple, educating clients on AAC and the therapy animal’s species and the therapy animal’s role, reviewing informed consent periodically, developing a safety protocol to handle potential issues, and continuously monitor their therapy animal for any signs of stress as well as continuous discussion and attention to verbal and non-verbal cues from clients in relation to human–animal interactions during session.

The author acknowledges that a formal consent process, as outlined previously, may not be feasible in all settings. In environments like prisons or other institutions where clients have limited autonomy, AAC practitioners face unique challenges. In these contexts, obtaining formal written consent might not be possible, but practitioners should still strive to conduct a verbal consent process whenever possible. This effort, though informal, is crucial in respecting the client’s dignity and autonomy within the constraints of the environment.

If a client expresses reluctance or unwillingness to participate in AAC but lacks the option to decline, the practitioner must prioritize the safety and well-being of both the client and the therapy animal. In such cases, the practitioner can take proactive measures to protect both parties, such as keeping the therapy animal close to them and physically distant from the client. If the therapy animal is a dog or cat, using a harness or leash can provide additional control and security. These precautions not only help prevent potential incidents but also subtly reinforce the concept of autonomy for the client—an experience that may be scarce in their current environment.

By carefully navigating these situations, AAC practitioners can demonstrate respect for the client’s boundaries and foster a sense of agency, even in restrictive settings. This approach not only upholds ethical standards but also enhances the therapeutic potential of AAC, ensuring that the intervention remains both safe and effective, regardless of the setting.

## 4. Discussion

Animal-assisted counseling (AAC) is increasingly recognized for its therapeutic benefits, particularly in enhancing the human–animal bond to improve client outcomes. However, as AAC gains popularity, it is critical to address the ethical, cultural, and practical implications of integrating therapy animals into mental health interventions. This discussion explores the complexities of AAC, particularly in diverse cultural contexts and restrictive environments, and highlights the importance of maintaining ethical standards and client autonomy throughout the therapeutic process.

### 4.1. Ethical and Cultural Dimensions in AAC

While AAC has demonstrated significant therapeutic potential, much of the existing literature overlooks the cultural, religious, and racial factors that influence client experiences with animals. This gap is particularly concerning given that people of color represent the global majority in America. For AAC to be truly inclusive and effective, practitioners must understand and respect these diverse perspectives.

Cultural and historical contexts play a crucial role in shaping clients’ relationships with animals. For instance, in some communities of color, animals have been used in aggressive and violent practices, leading to trauma that could affect a client’s willingness or ability to engage in AAC. Similarly, in various cultures and religions, animals may be revered, forbidden, or subject to specific rituals, all of which can influence a client’s comfort with AAC.

Given these complexities, AAC practitioners must engage in ongoing training to enhance their cultural competence and humility. This training should include a focus on understanding clients’ histories with animals, particularly any trauma they may have experienced, and evaluating the appropriateness of AAC for each client. These types of trainings and research in this area are also lacking within the field of AAC. By aligning training programs with AAC competencies, incorporating practical supervision, and encouraging further research and participation in literature, practitioners can develop and enhance the skills necessary to navigate these cultural considerations effectively.

### 4.2. Navigating Ethical Challenges in Restrictive Environments

This author worked, for five years, within a juvenile setting that had a facility canine and staff with no formal training at that time. From personal experience, this author witnessed healing and the positives of having a canine for some youth. On the flipside, the author also witnessed both staff and youth who had previous trauma with canines or varying beliefs about canine roles. This author witnessed how it could be dangerous for both the canine and the youth when the youth did not have autonomy to engage. Therefore, the author sees the importance of reviewing ethical challenges within these restrictive environments.

Animal-assisted interventions, including animal-assisted counseling, are becoming increasingly popular in restrictive environments such as prisons. Studies have shown improvement in mental health symptoms specifically with the integration of canine animal assisted interventions in prisons [37]. Statistics show that there are huge racial disparities within these environments in the United States. When breaking down statistics of those incarcerated in the United States, Black adults make up 38.9% of those incarcerated [38]. Hispanic adults make up 29.1% of those incarcerated [39]. Statistics show that youth of color face challenges such as overcriminalization. Youth of color make up over half of youth in the juvenile justice system. Specifically, Black and Hispanic youth represented 39.5% of the juvenile justice system in the United States in 2021 [40]. Data show further disproportion as, out of those youths who are placed within facilities such as detention centers or juvenile justice facilities and prisons, 41% of them are Black [40].

As previously stated, animals such as canines can be utilized within law enforcement and may cause trauma to those incarcerated. There are also generational experiences, religious beliefs, cultural beliefs, and perceptions of various animal species that are passed down to current generations. Bringing a therapy dog, at no fault of the animal, into a restrictive environment such as a prison can be triggering for some that are incarcerated. In environments where clients have limited autonomy, such as prisons or institutional settings, the ethical challenges of AAC become even more pronounced. In these settings, obtaining formal, written consent for AAC may not be possible, yet it remains crucial for practitioners to respect client autonomy and dignity as much as possible.

Ideally, an AAC practitioner would have an informed consent that would be signed by all people involved to participate in activities within these settings. If there is hesitancy that participants may not have had autonomy nor read through the consent, a verbal consent process, though informal, can help address this challenge by allowing clients to express their willingness to participate in AAC. There are also reward systems within environments such as a jail or prison in which incarcerated people may be given perks or benefits for participating in AAC and may not have wanting to opt out of participating because of this.

This can also be the case in academic or school settings. This author once had a participant in an animal-assisted activity from a college setting who was able to meet a requirement for their academic probation by attending my event. This author did not know that the event was being advertised, by a department on campus, as an incentive for learning about mental health. The student agreed to go to my event, but this student had a fear of rabbits. The author found that this fear was due to an incident in which a rabbit, who was not being well taken care of, bit them. The author had a therapy animal present who was a rabbit. Therefore, the author had to set limits for interaction and educate the client and the rest of the group without singling out the one participant, because the author did this for each participant that was present. The participant was engaged in discussion but was not near the therapy animal. If the author were to have assumed that everyone loves rabbits and that there would no issues, there would have been the potential of harm to either the client or the therapy animal. Therefore, the author will provide steps to navigate these situations.

Before introducing a therapy animal, a short verbal assessment of the client’s history, including any fears, allergies, or past trauma related to animals, can be helpful. When a client is unwilling or unable to participate but lacks the choice to decline, practitioners must prioritize the safety and well-being of both the client and the therapy animal. (1) Always supervise the therapy animal closely during sessions. (2) Keep the animal on a leash, harness, or in a carrier when appropriate to prevent it from approaching the client unexpectedly. (3) Regularly monitor the therapy animal’s behavior and stress levels. If the animal shows signs of distress or fatigue, end the session or provide the animal with a break. (4) Educate the client on how to interact appropriately with the therapy animal. This includes not touching the animal without permission, being gentle, and understanding the animal’s body language. (5) If there is a risk of the client becoming aggressive or overly excited, maintain a safe distance between the client and the animal. Consider using physical barriers if necessary. (6) Have a plan in place for removing the animal from the environment if the situation becomes unsafe. This includes a designated exit route and a calm, quick method to transport the animal to safety. By implementing these strategies, AAC practitioners can create a safe environment where both the client and therapy animal are protected, ensuring the therapeutic process remains positive and effective for all involved.

These precautions not only protect the client and animal but also model a form of autonomy for the client, offering them a sense of control and agency in settings where such experiences may be rare. This approach underscores the importance of maintaining ethical standards and client safety in all AAC interventions, regardless of the setting.

Though history has shown intense trauma to people of color involving the utilization of animals in traumatic ways, practitioners should also be cautious in only attempting to get consent from people of color, as all people can have their culture, societal experiences, and traumas. Practitioners should utilize this tool of consent with all clients regardless of their religion, race, ethnicity, and cultural beliefs. Practitioners should not assume their client’s religious and cultural beliefs. This requires practitioners to not be biased in limiting their cultural responsiveness to one group of people as that would defeat the purpose of practicing cultural humility and being culturally aware of themselves and others.

AAC involves fostering a mutually beneficial human–animal bond throughout the treatment process, with a focus on ensuring the welfare of both clients and therapy animals. The American Counseling Association (ACA), along with other professional organizations, emphasize the importance of understanding how cultural factors and history impact these interactions, assessing clients’ history of animal-related trauma and evaluating the suitability of both clients and animals for AAC. To prevent harm and enhance the effectiveness of AAC, practitioners should pursue specialized training, consider cultural differences, and implement protective measures. Research shows that aligning training programs with AAC competencies and combining them with practical supervision improves the skills necessary for successful AAC practice [15]. Additionally, practitioners must participate in ongoing training to stay updated on field developments and enhance cultural humility within the human–animal bond.

## 5. Conclusions

None of the information stated in this chapter should create a bias nor perpetuate stereotypes about specific groups of people, their cultures, and/or religions. The integration of animals into counseling presents a powerful opportunity to enhance therapeutic outcomes, but it also requires a careful, culturally informed, and ethically sound approach. AAC practitioners must remain vigilant in considering the diverse cultural backgrounds of their clients and the specific challenges posed by restrictive environments. By doing so, they can ensure that AAC interventions are not only effective but also respectful, inclusive, and safe for all involved.

As the field of AAC continues to evolve, it is imperative that research and practice keep pace with the growing understanding of cultural and ethical complexities. Access to animal-assisted interventions may be limited, which can also limit research. Clients from lower socioeconomic backgrounds might face barriers such as affordability, availability of services, or access to facilities where AAC is offered. Practitioners and researchers can fill this gap by providing pro-bono work and/or applying for grants to organize activities and research. By prioritizing cultural humility, ongoing training, and client autonomy, AAC practitioners can contribute to a more equitable and effective therapeutic landscape.

If following ethical, professional, and competency standards, AAC practitioners should assess whether services are appropriate for each client. These practitioners should seek intensive training specific to working with animals in mental health settings and ensure that they follow AAC competencies and their professional ethics to avoid unwanted injuries and harm to the animal or the clients. Practitioners should attend trainings to remain aware of changes within the field and to increase cultural humility within the human–animal bond. AAC practitioners can create an interest form and conduct an initial interview to assess client appropriateness for AAC along with an informed consent that helps clients remain aware of the process of AAC. When in restrictive settings, practitioners can alter their approach to the needs of their clients, as all people and species of animals are different.

## Figures and Tables

**Figure 1 animals-14-02496-f001:**
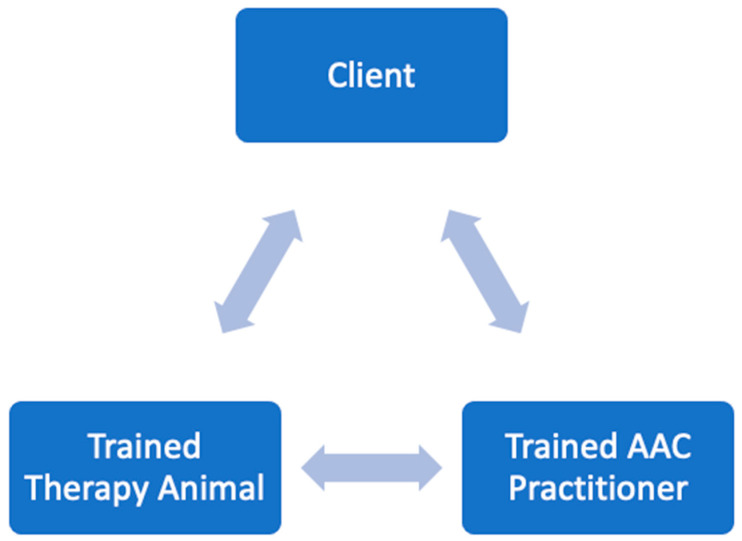
AAC Therapeutic Relationship.

**Figure 2 animals-14-02496-f002:**
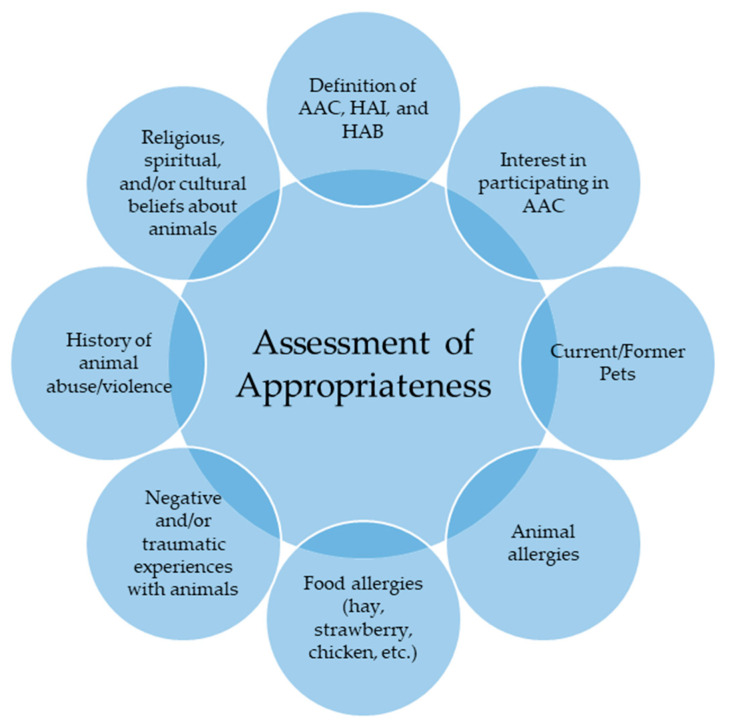
Assessment of Appropriateness.

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
