# Peer review of "Assessing Cultural, Religious, and Trauma Influences in Human-Animal Interactions for Effective Animal-Assisted Counseling"

_animals, 2024, doi:10.3390/ani14172496_

Round 1

Reviewer 1 Report

Comments and Suggestions for Authors

The point of this paper is good. However, the presentation needs to be improved.

Race is a socially constructed notion. How can you argue that "the impact of racial, social, and cultural influences on AAC" (line 22-23, 57,  and other places)? In fact, you never analyzed racial influence on AAC in your paper. If you want to argue that race influences AAC, please provide data to back up your argument. This is the weakest point of this paper.

Also, the writing needs to be improved. For example, section 2.1 does not flow smoothly.  

Other points include;

Muslim religion = Islam (Line 137); "By" instead of "due to" (Liine 118); "Indian American reservation (Line 152)

The Discussion section (conclusion) is not strong. Please make it more stronger since this is a concept paper.

I believe this concept paper makes a good point. It just needs to be polished.

Comments on the Quality of English Language

As stated above, this paper needs to be polished.

Author Response

Thank you for taking the time to review this manuscript. Please find the detailed responses below and the corresponding revisions in the re-submitted file (please see attached).

Does the introduction provide sufficient background and include all relevant references? Can be improved. Agreed.

Are the results clearly presented? Must be improved. Agreed

Are the conclusions supported by the results? Must be improved. Agreed

Comment 1: The point of this paper is good. However, the presentation needs to be improved.

Response 1: Thank you for pointing this out. Throughout the paper, you can see that I edited it to flow better and organized topics (Was not sure whether to make the whole paper red)

Comment 2: Race is a socially constructed notion. How can you argue that "the impact of racial, social, and cultural influences on AAC" (line 22-23, 57,  and other places)? In fact, you never analyzed racial influence on AAC in your paper. If you want to argue that race influences AAC, please provide data to back up your argument. This is the weakest point of this paper.

Response 2: I agree with this comment. Thank you. I have changed this throughout the paper and improved each section.(Was not sure whether to make the whole paper red)

Comment 3: Also, the writing needs to be improved. For example, section 2.1 does not flow smoothly.  

Response 4: Thank you. I have completely revised Section 2.1. This can be viewed on pages 4-6.

Comment 4: Other points include;

Muslim religion = Islam (Line 137); "By" instead of "due to" (Liine 118); "Indian American reservation (Line 152)

Response 4: Thank you, I have revised this and you can see it throughout page 5 starting at line 190 and ending on line 219.

Comment 5: The Discussion section (conclusion) is not strong. Please make it more stronger since this is a concept paper.

Response 5: Thank you. I agree. You can see the revised Discussion section and an added Conclusion section on pages 9-12. 

Comment 6: I believe this concept paper makes a good point. It just needs to be polished.

Response 6: Thank you. I wanted to be bring awareness to this topic and greatly appreciated your feedback. 

Reviewer 2 Report

Comments and Suggestions for Authors

-I would suggest finding a different word other than "use" in the opening sentence. "Integration" would be preferred. 

-There are empirically supported Professional Competencies for the Inclusion of Therapy and Facility animals available from the Association of Animal-Assisted Intervention Professionals. To best reflect the state of the field, I'd suggest incorporating a mention of these in the paper. 

-Claims in the introduction section could be enhanced with more empirical citations.

-Line 105 appears to be missing a word. 

-should "animal assisted" be hyphenated throughout? 

-line 129 should be re-worded: perhaps using "AAC professionals" rather than "counseling counselors"

-line 162: what does is mean to the author to be "certified" in animal-assisted counseling. Is this referring to AAAIP's AAIS certification, or another certification in the field? 

-The topics of trauma, religion, and culture are huge. I would suggest further building out the section of the paper. 

-The discussion section could be further elaborated on. 

-I'm unclear as to why "definitions" is included in figure 2. More explanation on that aspect is merited. 

Comments on the Quality of English Language

minor revisions as suggested above 

Author Response

Thank you for your review. Please find the detailed responses below and the corresponding revisions in the re-submitted files (Please see the attached). 

Does the introduction provide sufficient background and include all relevant references? Can be improved. Agreed.

Are the results clearly presented? Can be improved. Agreed

Are the conclusions supported by the results? Can be improved. Agreed

Comment 1: I would suggest finding a different word other than "use" in the opening sentence. "Integration" would be preferred. 

Response 1: I have changed this word throughout the paper. This was a mistake on my part. I usually utilize the word "work with" or "integrate" when discussing a therapy animal's work.

Comment 2: There are empirically supported Professional Competencies for the Inclusion of Therapy and Facility animals available from the Association of Animal-Assisted Intervention Professionals. To best reflect the state of the field, I'd suggest incorporating a mention of these in the paper. 

Response 2: Thank you for this reminder. I was focused on America's competencies which are not very inclusive in the first place. AAIP was a great resource. Utilized this resource throughout when discussing sets of competencies. Starting in the abstract. 

Comment 3: Claims in the introduction section could be enhanced with more empirical citations.

Response 3: Completely revised the introduction. Highlighted in red. 

Comment 4: Line 105 appears to be missing a word. 

Response 4: I could not identify what this missing word was. I did go through the paper and ensured that I filled the gaps and corrected grammatical mistakes. 

Comment 5: should "animal assisted" be hyphenated throughout? 

Response 5: I hyphenated these words throughout (was not sure if I needed to make the entire paper red for the changes that I made throughout)

Comment 6: line 129 should be re-worded: perhaps using "AAC professionals" rather than "counseling counselors"

Response 6: made this change throughout the paper. 

Comment 7: line 162: what does is mean to the author to be "certified" in animal-assisted counseling. Is this referring to AAAIP's AAIS certification, or another certification in the field? 

Response 7: made the change to "trained" throughout as the certification process may look different in various places in the world. I am trained and certified through a training program here in the United States but that may look different for others. I think I explained the type of training that AAC practitioners can receive on page 3, lines 109-124.

Comment 8: The topics of trauma, religion, and culture are huge. I would suggest further building out the section of the paper. 

Response 8: I edited this throughout the paper but specifically section 2.1 was completely revised as seen on pages 4-6.

Comment 9: The discussion section could be further elaborated on. 

Response 9: Completely revised the discussion and added a conclusion. see pages 9-12.

Comment 10: I'm unclear as to why "definitions" is included in figure 2. More explanation on that aspect is merited. 

Response 10: Thank you for this feedback. On line 286, I gave a brief message about what this is for. I discuss education clients throughout and this is part of that. 

Reviewer 3 Report

Comments and Suggestions for Authors

You will need to position yourself if you are going to discuss race and culture AND I hope you are of a race and culture that you discuss. otherwise it is a little othering.

line 109- who is the we?  written history shows?

line 146 add alleged before superiors?

line 149- who is the we here?

202 an example of what might constitute either cultural responsiveness or cultural humility would strengthen

223: I think you should not be too prescription focussed in this. It will change with cultures and beliefs as to how it might need to be included in health and wellness of client and animal

223-233 this is a very western approach to everything. Perhaps something to think about if again covering diversity of culture and race

I believe some more linking to cultural responsiveness and cultural humility is needed. A discussion of when consent is not possible- which occurs for many minority races in prisons for example would strengthen

Comments on the Quality of English Language

Was fine

Author Response

Thank you for your review. Please find the detailed responses below and the corresponding revisions in the re-submitted files (Please see the attached). 

Does the introduction provide sufficient background and include all relevant references? Yes.I completely edited this section. I thought it needed some improvement.

Comment 1: You will need to position yourself if you are going to discuss race and culture AND I hope you are of a race and culture that you discuss. otherwise it is a little othering.

Response 1: Lines 37-42 are an introduction that I added for myself. Not sure if this is appropriate but I do agree with you that people need to know that I am apart of multiple marginalized groups. 

Comment 2: line 109- who is the we?  written history shows?

Response 2: Thank you. I changed this throughout the paper (was not sure if I needed to make the entire paper red or state the parts in which I changed "we"). 

Comment 3: line 146 add alleged before superiors?

Response 3: See line 182 for this change. 

Comment 4: line 149- who is the we here?

Response 4: Thank you. I changed this throughout the paper

Comment 5: 202 an example of what might constitute either cultural responsiveness or cultural humility would strengthen

Response 5: Thank you. In line 196, I discussed that cultural responsiveness should be utilized with all groups. Throughout the paper, I added discussion of not assuming people's cultural beliefs and the importance of utilizing cultural humility at all times, not just during assessment. 

Comment 6: 223: I think you should not be too prescription focussed in this. It will change with cultures and beliefs as to how it might need to be included in health and wellness of client and animal

Response 6: Starting at line 342, I begin to discuss what to do if practitioners are not able to follow this type of process. I completely understand what you are saying as I have worked with marginalized people my whole career and am a marginalized person myself. I hope that this response was correctly interpreted by me. Thank you. 

Comment 7: 223-233 this is a very western approach to everything. Perhaps something to think about if again covering diversity of culture and race

Response 7: Thank you. Yes, I agree. I think that since AAC is new in general and then adding in that most of the research is done by white researchers in white communities, there is a lack of what is appropriate for communities of color and marginalized folks and people who have different beliefs and norms than the colonized, westernized perceptions of some people in America. I added a lot of information throughout the paper and added a lot of discussion points on this in the Discussions section. Again, thank you very much. Being the first person in the AAC world specifically, that I know of, to write about this was a challenge. 

Comment 8: I believe some more linking to cultural responsiveness and cultural humility is needed. A discussion of when consent is not possible- which occurs for many minority races in prisons for example would strengthen

Response 8: Thank you. I worked in a juvenile justice facility for many years. I added this experience in section 4.2 and discussed how to navigate this. I really appreciated your feedback. 

Round 2

Reviewer 1 Report

Comments and Suggestions for Authors

I appreciate the revision.

What are the differences between your argument and what AAAIP is stating? Are you proposing a more robust examination of cultural and racial factors in use of AAC? I am still not clear on that.

Comments on the Quality of English Language

Lines 48-50 -- Too long? The point is not clear

Line 66 - "implications"?

Line 68 - "societies"?

Author Response

Please see revised manuscript attached.

Comment 1: I appreciate the revision.

Response 1: I appreciate your feedback!

Comment 2: What are the differences between your argument and what AAAIP is stating? Are you proposing a more robust examination of cultural and racial factors in use of AAC? I am still not clear on that.

Response 2: Yes, I don't like how things can sometimes be performative or a "check the box." I added this explanation on lines 30-37. Thank you for this question. 

Comment 3: Lines 48-50 -- Too long? The point is not clear Response 3: Thank you. Edited this and made it clear. Lines 53-55. 

Comment 4: Line 66 - "implications"?

Response 4: Changed this whole sentence. Reading it out loud again sounded weird. Thank you. Change is on line 71. 

Comment 5: Line 68 - "societies"?

Response 5: Changed my grammar here. Line 72 shows changes. Thank you 

Reviewer 2 Report

Comments and Suggestions for Authors

The author did a great job addressing my previous comments and has significantly added to the paper. 

Author Response

Comment 1: The author did a great job addressing my previous comments and has significantly added to the paper. 

Response 1: Thank you. Are there are any other edits to be made in order for the review report to be completed?